# Rotordynamic Analysis and Operating Test of an Externally Pressurized Gas Bearing Turbo Expander for Cryogenic Applications

Donghyun Lee *, Hyungsoo Lim, Byungock Kim, Byungchan Jeon and Junyoung Park

Korea Institute of Machinery and Materials, Gajeongbuk-ro, Yuseong-gu, Daejeon 34103, Republic of Korea; limbo999@kimm.re.kr (H.L.); kbo2612@kimm.re.kr (B.K.); arice1008@kimm.re.kr (B.J.); jypark@kimm.re.kr (J.P.)
* Correspondence: donghyun2@kimm.re.kr; Tel.: +82-42-868-7662

**Abstract:** This study designed an externally pressurized bearing and analyzed the rotordynamics of a turbo expander for a hydrogen liquefaction plant. The turbo expander, comprising a turbine and compressor wheel assembled to a shaft, lowered the temperature of the helium refrigerant. Its rated speed was 75,000 rpm, and an externally pressurized gas bearing was selected to support the rotor. Pressurized helium was used as the lubricant for the bearing operation. To design the rotor–bearing system, we conducted a bearing performance analysis and rotordynamic characteristic prediction using the developed numerical model. We calculated the bearing stiffness and flow rate of the bearing gas for various feed parameters and selected the appropriate orifice diameter for maximum stiffness. The predicted Campbell diagram showed that the system had a sufficient separation margin with the critical speed, and the predicted critical speed correlated well with the nonlinear orbit simulation. A successful operation was achieved with the manufactured turbo expander within the rated speed. The shaft vibration was monitored during the operation test, and the test results revealed two critical speeds below the rated speed, as predicted by the analytical model. In addition, the shaft vibration was maintained at <3 μm.

**Keywords:** turbo expander; externally pressurized gas bearing; hydrogen liquefaction plant

## 1. Introduction

Climate change, caused by the increase in greenhouse gas emissions, is currently the most pressing global challenge that needs to be addressed. To limit the global average temperature increase to less than 2 °C, the Paris Agreement was established in 2015 as a legally binding international treaty. In this regard, hydrogen has emerged as an attractive solution attention to mitigate climate change as it serves as a carbon-free fuel in gas turbines, internal combustion engines, and fuel cells. Moreover, hydrogen can be utilized as an effective energy storage source for renewable energy. In energy storage devices, hydro-gen can store more energy for a longer period of time without capacity degradation compared to conventional batteries. Because of its great potential for energy storage, hydrogen is gaining traction as a commercial energy source in many countries. Thus, a system for the storage and transportation of large volumes of hydrogen should be established for massive deployment. Among various hydrogen storage systems, liquid hydrogen is considered a promising solution owing to its high density and safety resulting from low-pressure storage. Consequently, several studies have focused on hydrogen liquefaction processes and the development of core technologies for hydrogen liquefaction plants.

In a hydrogen liquefaction plant, a turbo expander is used as the core component to lower the refrigerant temperature. The turbo expander used in hydrogen liquefaction plants is designed to operate under extreme conditions of cryogenic temperature, speed, and pressure. Under these operating conditions, conventional oil bearings are not feasible;

thus, oil-free bearings are typically adopted for turbo expanders. Among the various oil-free bearings, externally pressurized gas bearings are widely adopted in turbo expanders owing to their various advantages.

Externally pressurized gas bearings are operated by supplying pressurized gas to the bearing, resulting in a superior load capacity and precise operation compared to conventional bearings. Owing to these benefits, externally pressurized bearings are advantageous for various applications, and thorough investigations have been conducted in this regard. Additionally, Fleming et al. performed a theoretical study on a pressurized gas bearing with two feeding lines [1]. They showed that the stability decreases with increasing recess volume and with the pressure ratio under the condition of a low bearing number. In a subsequent study, the authors measured the dynamic stiffness and damping of a pressurized gas bearing and reported that the measured stiffness was proportional to the supply pressure [2]. Moreover, Chen et al. theoretically investigated the effect of geometric parameters on the stiffness of a pressurized gas bearing and demonstrated the credibility of the analytical model by comparing predictions with measured data [3]. Furthermore, Belforte et al. conducted a numerical investigation to optimize aerostatic bearings for high-speed spindle applications. They demonstrated that the predicted stiffness agreed well with the test results [4]. Furthermore, Xiao et al. proposed the optimum design parameters for an aerostatic microbearing based on a parametric study, revealing that the hydrodynamic effects are considerable compared to the hydrostatic effects in bearings under high-speed and large-eccentricity conditions [5]. Otsu et al. conducted a theoretical investigation on the performance of an aerostatic bearing [6]. They considered the inertial effects of the lubricant and showed that a bearing with a compound restrictor had a larger stiffness than that with an inherently compensated restrictor. Additionally, the authors validated their predictions through comparison with their experimental data. Yang et al. compared three types of orifice arrays and showed the existence of optimum restriction parameters [7]. Chen et al. compared the static and dynamic performances of an orifice with inherently compensated bearings [8]. They reported that the load capacity and stability of an orifice-compensated bearing are superior to those of an inherently compensated bearing. Additionally, Loa et al. presented a theoretical study on pressurized gas bearings for high-speed spindle applications [9]. They demonstrated that the optimum bearing clearance for the maximum stiffness is not identical to that required for the maximum load capacity. Moreover, advanced methods were employed to enhance the accuracy of bearing performance prediction. Song et al. calculated the discharge coefficients of pressurized air bearings by taking into account the rotating speed and varying film thickness [10]. Zhang et al. proposed a numerical method that combines boundary-layer equations with the Reynolds equation to predict the discharge coefficients of inherent orifice restrictors [11]. Guo et al. conducted performance prediction of pressurized air bearings while considering structural deformation caused by pressurized air supply [12]. They demonstrated that an increase in thrust collar thickness led to an increase in bearing stiffness. Gao et al. performed design optimization using a multi-objective optimization genetic algorithm to maximize stiffness and showed a 38.5% increase in the stiffness of the selected bearing design [13].

In addition to the aforementioned studies on orifice restrictor bearings, studies on slot restrictor bearings have also been conducted for externally pressurized gas bearings. Rowe and Stout calculated the load capacities and flow rates of single- and double-row slot restrictor bearings [14]. They showed that an optimum gauge pressure ratio corresponds to the maximum load. Stout theoretically compared the load capacity between orifice and slot restrictor bearings and showed that the orifice restrictor bearing has a greater load capacity than the slot restrictor under small eccentricity conditions [15]. Moreover, Yoshimoto et al. investigated the characteristics of aerostatic bearings with circular slot restrictors [16]. They reported that bearings with circular slots have a load capacity comparable to that of bearings with discrete slots and showed that the predicted load capacity was in good agreement with the test results. In a subsequent study, Yoshimoto et al. predicted the stability of a slot restrictor bearing by considering the energy loss at the outlet of the

restrictors [17]. They suggested an energy loss coefficient based on experimental data, and their theoretical model accurately predicted the bearing stability. Furthermore, Tawfik and Stout proposed optimized design parameters for minimizing power consumption [18]. Park et al. proposed a bearing with a nonuniform slot and demonstrated the superior stability of the proposed bearing [19]. Kim et al. analyzed the dynamic characteristics of externally pressurized air bearings; reportedly, the feed parameter that maximizes the bearing stiffness is not identical to that required for optimizing damping [20].

Owing to their numerous advantages, pressurized gas bearings have been successfully applied to various industrial rotating types of machinery. However, only a few studies have focused on the application of pressurized gas bearings in cryogenic turbo expanders. This study focuses on the development of a turbo expander for a hydrogen liquefaction plant and presents a design approach for a rotor-bearing system with pressurized gas bearings. In particular, we performed bearing performance prediction and rotordynamic analysis of the turbo expander. For the rotordynamic analysis, linear and nonlinear analyses were conducted to predict the rotor–bearing characteristics more accurately. In addition, to demonstrate the viability of the designed turbo expander, we conducted operating tests up to the rated speed.

## 2. Materials and Methods

### 2.1. Turbo Expander Design

Figure 1 shows the configuration of the turbo expander for a 0.5 tons per day (TPD) hydrogen liquefaction plant under development [21]. In this process, helium is used as the refrigerant, and liquefied hydrogen is produced by absorbing cold heat from helium. A designed turbo expander is used to lower the helium temperature. The expansion ratio of the expander is 1.6, and the temperatures of the inlet and outlet gas are 49.1 K and 42.8 K, respectively. The rated speed of the turbo expander is 75,000 rpm, and the rated power is 6 kW. In the designed turbo expander, the turbine and compressor impeller are assembled on both sides of the shaft, as shown in Figure 1. A compressor is used in the expander to control the speed. The shaft is supported by two radial bearings near the compressor and turbine impeller. To support the axial motion of the rotor, a pair of thrust bearings are installed on both sides of the thrust collar. The shaft is installed vertically; therefore, its weight is supported by a thrust bearing. For additional information regarding the performance of turbo expanders, refer to [21].

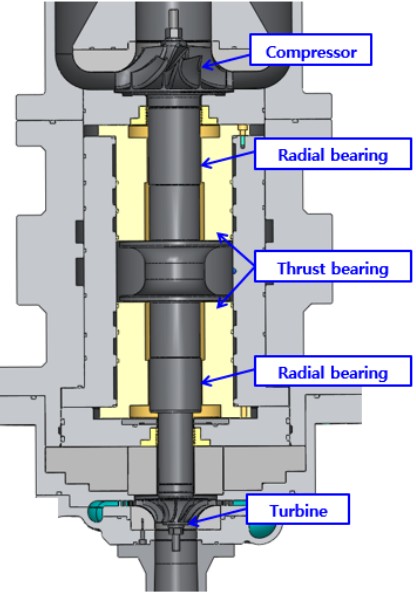

**Figure 1.** Schematic of the turbo expander.

Figure 2 presents the schematic of a radial bearing. Helium, which is the working fluid of the expander, is utilized as the bearing gas. Thus, the bearings are operated by supplying pressurized helium. The gas supply holes are created in two rows, and 16 supply holes are created in the circumferential direction in each row. The axial location of the supply hole is at a bearing length of 0.25, and orifice restrictors are fabricated as flow control devices. The other design parameters of the radial bearings and rotors are listed in Table 1.

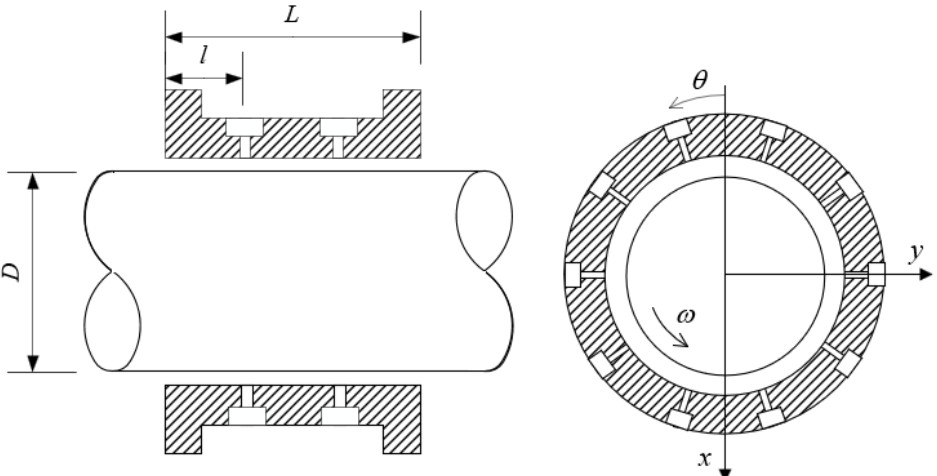

**Figure 2.** Illustration of the pressurized gas bearing.

**Table 1.** Design parameters for radial bearing and rotor.

| Properties | Symbol | Unit | Value |
|---|---|---|---|
| Diameter | $D$ | mm | 33 |
| Length | $L$ | mm | 60 |
| Bearing clearance | $C$ | mm | 0.03 |
| Rotating speed | $\omega$ | rpm | 75,000 |
| Discharge coefficient | $C_d$ | - | 0.8 |
| Supply pressure | $p_s$ | bar | 8.5 |
| Supply temperature | $T_s$ | °C | 20 |
| Number of orifice row | - | EA | 2 |
| Number of orifice per row | $N$ | EA | 16 |
| Rotor mass | $M$ | kg | 1.59 |
| Rotor polar moment of inertia | $Ip$ | kg·mm$^2$ | 413 |
| Rotor translational moment of inertia | $It$ | kg·mm$^2$ | 8137 |

## 2.2. Theoretical Model

### 2.2.1. Bearing Performance Analysis

Following the typical assumptions for gas lubrication, the bearing pressure can be calculated using the Reynolds equation for a compressive fluid, as shown in Equation (1).

$$\frac{\partial}{\partial \theta}\left(\Lambda PH - PH^3 \frac{\partial P}{\partial \theta}\right) + \frac{\partial}{\partial Z}\left(-PH^3 \frac{\partial P}{\partial Z}\right) + 2\Lambda\nu\frac{\partial}{\partial \tau}(PH) = \frac{\dot{M}_s}{\Delta\theta\Delta Z}. \tag{1}$$

The dimensionless variables used in Equation (1) are defined in Equation (2).

$$\theta = \frac{x}{R}, Z = \frac{z}{R}, P = \frac{p}{p_a}, H = \frac{h}{C}, \tau = \omega_s t, \Lambda = \frac{6\mu\omega}{p_a}\left(\frac{R}{C}\right)^2, \nu = \frac{\omega_s}{\omega}, \dot{M}_S = \frac{12\mu R_g T \dot{m}_s}{p_a{}^2 C^3}, \tag{2}$$

where $C$ is the bearing clearance, $R$ is the bearing radius, $h$ is the film thickness, $\mu$ is the viscosity of the lubricant, $p_a$ is the ambient pressure, $\omega$ is the rotational speed of the shaft, $\omega_s$ is the excitation frequency, $R_g$ is the gas constant, and $T$ is the temperature of the lubricant. The dimensionless flow rate supplied through the orifice is calculated assuming an isentropic flow and can be expressed using Equation (3).

$$\dot{M}_S = \Gamma_s P_s H \Phi. \tag{3}$$

The feed parameter, $\Gamma_s$, is defined as follows:

$$\Gamma_s = \frac{12\mu C_d A_0 \sqrt{R_g T}}{p_a C^3}. \tag{4}$$

The orifice function is defined as follows:

$$
\begin{aligned}
\Phi &= \left[ \frac{2k}{k-1} \left( \left(\frac{P}{P_s}\right)^{2/k} - \left(\frac{P}{P_s}\right)^{(k+1)/k} \right) \right]^{1/2}, \ \frac{P}{P_s} > \left(\frac{2}{k+1}\right)^{\frac{k}{(k-1)}} \\
\Phi &= \left(\frac{2k}{k+1}\right)^{1/2} \left(\frac{2}{k+1}\right)^{1/(k-1)}, \ \frac{P}{P_s} < \left(\frac{2}{k+1}\right)^{\frac{k}{(k-1)}}
\end{aligned}
, \tag{5}
$$

where $P_s$ is the nondimensional supply pressure, $C_d$ is the orifice discharge coefficient, $A_o$ is the orifice area, and $k$ is the specific heat ratio of the bearing gas.

The static bearing performance was predicted by solving the aforementioned governing equations, and the bearing stiffness and damping were calculated on the basis of the governing equations derived using the perturbation method derived from Equation (1) [22,23].

### 2.2.2. Rotordynamic Analysis

Figure 3 shows a linear rotor dynamics analysis model for predicting the vibration characteristics of a turbo expander. The rotor–bearing system is modeled as a multiple-degrees-of-freedom system, and the equation of motion of the rotor–bearing system can be written as

$$[M]\ddot{q}(t) + [C]\dot{q}(t) + [K]q(t) = f(t), \tag{6}$$

where $[M]$, $[C]$, and $[K]$ denote the generalized mass, damping, and stiffness matrices, respectively. The rotor was modeled using Euler–Bernoulli beam elements, whereas the turbine and compressor impellers were modeled with equivalent inertia. The stiffness and damping of the bearing were calculated using linearized perturbation theory.

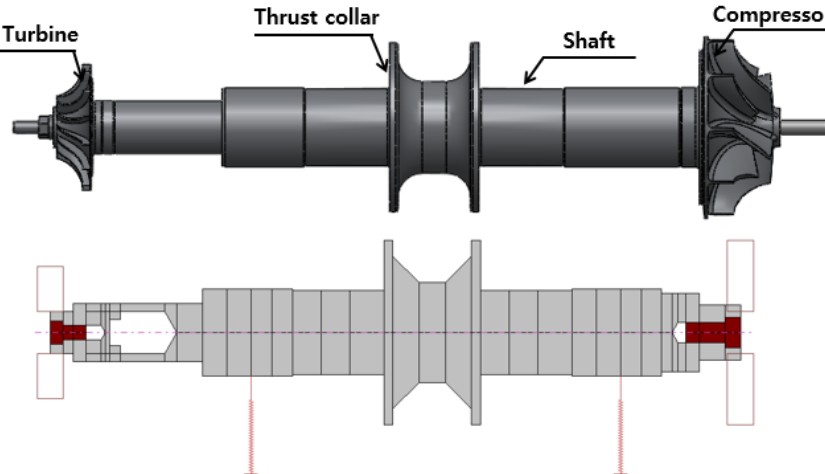

**Figure 3.** Rotordynamic analysis model for turbo expander.

In the nonlinear orbit simulation, the rotor was assumed to be a rigid body. Figure 4 shows the coordinated system and variables describing the rotor motions. In the analysis, motions with four degrees of freedom were considered, determined by solving the following equations:

$$
\begin{aligned}
m\ddot{X} &= F_{X\_B} + F_{X\_U} \\
m\ddot{Y} &= F_{Y\_B} + F_{Y\_U} \\
I_T\ddot{\xi} + I_P\omega\dot{\psi} &= M_{\xi\_B} + M_{\xi\_U} \\
I_T\ddot{\psi} - I_P\omega\dot{\xi} &= M_{\psi\_B} + M_{\psi\_U}
\end{aligned}
\tag{7}
$$

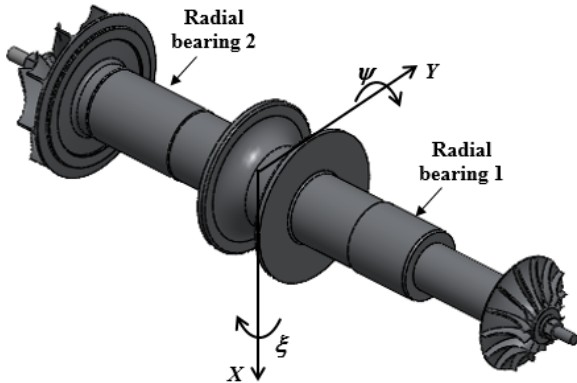

**Figure 4.** Coordinate system of rotor motion for nonlinear orbit simulation.

In Equation (7), $m$ is the rotor mass, $X$ and $Y$ are the rotor radial motions, and $\xi$ and $\psi$ are the rotor conical motions. $I_T$ is the translational moment of inertia of the rotor, and $I_p$ is the polar moment of inertia of the rotor. In this study, $F_{X\_U}$, $F_{Y\_U}$, $M_{\xi\_U}$, and $M_{\psi\_U}$ are the forces and moments induced by the imbalanced mass of the rotor, respectively. It is assumed that the mass imbalances are located at the turbine and compressor impellers; therefore, they are given by

$$
\begin{aligned}
\left\{ \begin{array}{c} F_{X\_U} \\ F_{Y\_U} \end{array} \right\} &= \left\{ \begin{array}{c} u_1\cos\omega t + u_2\cos(\omega t + \phi_p) \\ u_1\sin\omega t + u_2\sin(\omega t + \phi_p) \end{array} \right\} \\
\left\{ \begin{array}{c} M_{\xi\_U} \\ M_{\psi\_U} \end{array} \right\} &= \left\{ \begin{array}{c} -u_1\sin\omega t \cdot l_1 + u_2\sin(\omega t + \phi_p)\cdot l_2 \\ u_1\cos\omega t \cdot l_1 - u_2\cos(\omega t + \phi_p)\cdot l_2 \end{array} \right\}
\end{aligned}
\tag{8}
$$

In Equation (8), $u_1$ and $u_2$ represent the number of mass imbalances located at the turbine and compressor impellers, respectively, $l_1$ and $l_2$ are the axial distances between the center of mass of the rotor and the imbalance locations, respectively, and $\phi_p$ is the angle between the imbalances at the turbine and compressor impellers. $F_{X\_B}$, $F_{Y\_B}$, $M_{\xi\_B}$, and $M_{\psi\_B}$ are the forces and moments induced by the radial bearings. The forces and moments are calculated using the following equations:

$$
\begin{aligned}
\left\{ \begin{array}{c} F_{X\_B} \\ F_{Y\_B} \end{array} \right\} &= \left\{ \begin{array}{c} \iint p_1\cos\theta r d\theta dz + \iint p_2\cos\theta r d\theta dz \\ \iint p_1\sin\theta r d\theta dz + \iint p_2\sin\theta r d\theta dz \end{array} \right\} \\
\left\{ \begin{array}{c} M_{\xi\_B} \\ M_{\psi\_B} \end{array} \right\} &= \left\{ \begin{array}{c} -\iint z p_1\sin\theta r d\theta dz - \iint z p_2\sin\theta r d\theta dz \\ \iint z p_1\cos\theta r z d\theta dz + \iint z p_2\cos\theta r d\theta dz \end{array} \right\}
\end{aligned}
\tag{9}
$$

In the above equations, $p_1$ and $p_2$ are the pressures in the radial bearings at the turbine and compressor side, as shown in Figure 4. In the nonlinear orbit simulation, the bearing pressure was calculated by solving Equation (1) in the time domain.

## 3. Results and Discussion

### 3.1. Validation of Theoretical Model

To validate the bearing performance analysis model developed in this study, the predicted results were compared with the data available in the literature [16]. Figure 5

shows the predicted dimensionless stiffness according to the feed parameters for the bearings with $L/D = 1$, $\Lambda = 1$, and $\varepsilon = 0.1$. The dimensionless stiffness is defined as $K = kC/p_a/L/D$. The analysis was performed for cases with supply pressures of 0.5 and 0.7 MPa. The results predicted in the reference study [16] and those calculated using the program in the current study are presented herein. As shown in the figure, the results of the reference study [16] and those in the current study were within 5% of each other.

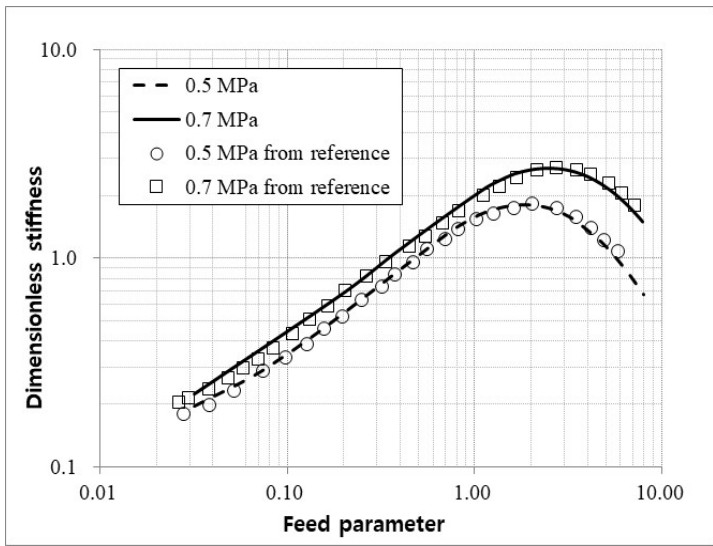

**Figure 5.** Dimensionless stiffness for pressurized gas bearing reported in the literature [20].

### 3.2. Bearing Design and Performance Analysis

To select the appropriate orifice diameter of the radial bearing, the bearing stiffness and flow rate were calculated according to the feed parameters, as described in Table 1. The analysis was conducted at a rated speed of 75,000 rpm and a concentric condition of the rotor because the turbo expander was operated in the vertical direction, as shown in Figure 1. Pressurized helium (8.5 bar) was used as the bearing gas.

Figure 6 plots the bearing stiffness and flow rate as a function of the feed parameters. The bearing stiffness in this figure is $k_{xx}$, as defined in the coordinate system shown in Figure 2. The stiffness is calculated on the synchronous condition with rotating speed. Because the performance analysis was conducted under concentric conditions, the vertical and horizontal stiffness values were identical. As shown in the figure, the predicted results revealed a feed parameter that maximized the stiffness of the bearing. In this analysis, the maximum stiffness was predicted to be approximate to a feed parameter of 1.9. However, due to an orifice machining issue, we chose to select the orifice diameter at the feed parameter of 1.43, even though the stiffness at this point was 3% lower than the maximum value. In addition, the flow rate of the bearing gas increased with the feed parameters, as shown in Figure 6b, and the flow rate was predicted to be 0.54 g/s at the selected feed parameter.

Figure 7 shows the predicted load-carrying capacity of the radial bearing as a function of the eccentricity ratio at a rated speed of 75,000 rpm. The load-carrying capacity increased almost linearly, and the designed bearing was predicted to exhibit a load capacity of 180 N at an eccentricity ratio of 0.5. Accordingly, the designed bearing was predicted to support radial loads up to 180 N at the rated speed.

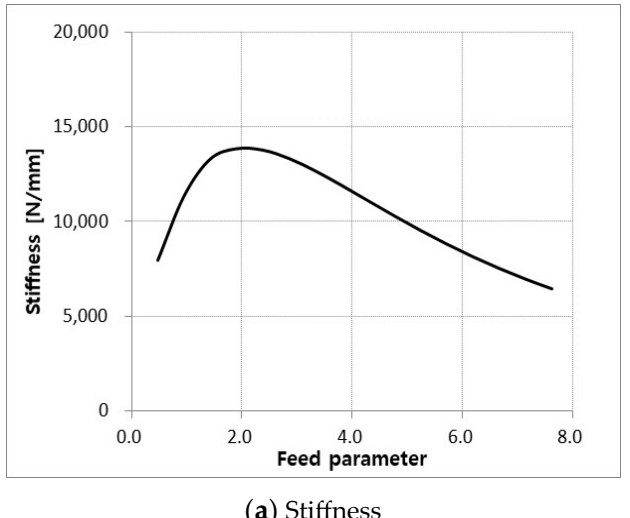

(**a**) Stiffness

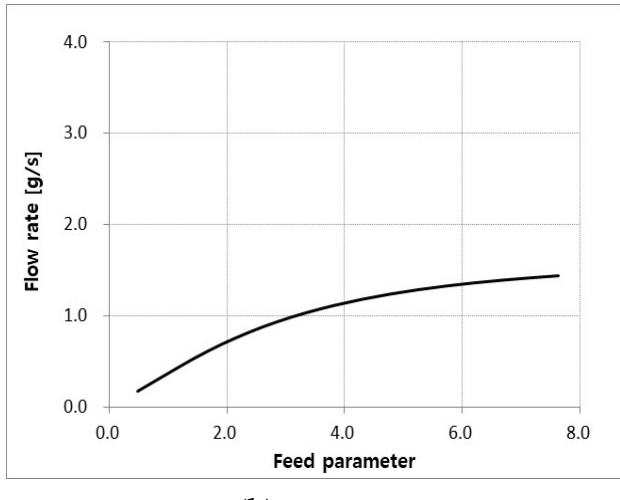

(**b**) Flow rate

**Figure 6.** Stiffness and flow rate for feed parameters ($\omega$ = 75,000 rpm).

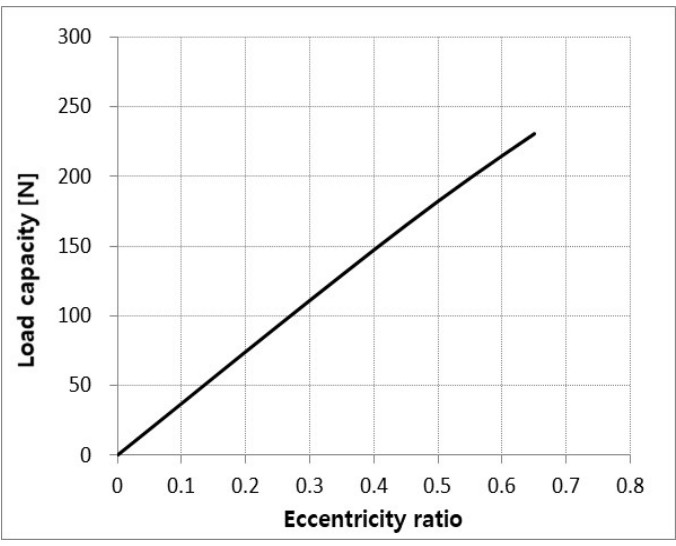

**Figure 7.** Load capacity versus eccentricity ratio ($\omega$ = 75,000 rpm).

Figure 8 plots the predicted bearing stiffness and damping with respect to rotor speed. The stiffness and damping were calculated in synchronous conditions. The direct stiffness and damping in the vertical direction ($k_{xx}$, $c_{xx}$) were identical to those in the horizontal direction ($k_{yy}$, $c_{yy}$) owing to the concentric operation of the rotor. Therefore, the rotor–bearing system of the turbo expander is expected to exhibit the characteristics of an isotropic rotor. As the rotor speed increased, the cross-coupled stiffness increased, indicating that the hydrodynamic effects dominated over the hydrostatic effects at high speeds. Additionally, cross-coupled stiffness appeared in the form of skew symmetry; thus, the possibility of instability should be verified under high-speed conditions. However, the direct stiffness significantly exceeded the cross-coupled stiffness at all speeds. For the damping coefficient, the direct damping decreased, whereas the cross-coupled damping increased with the rotor speed, as shown in Figure 8b.

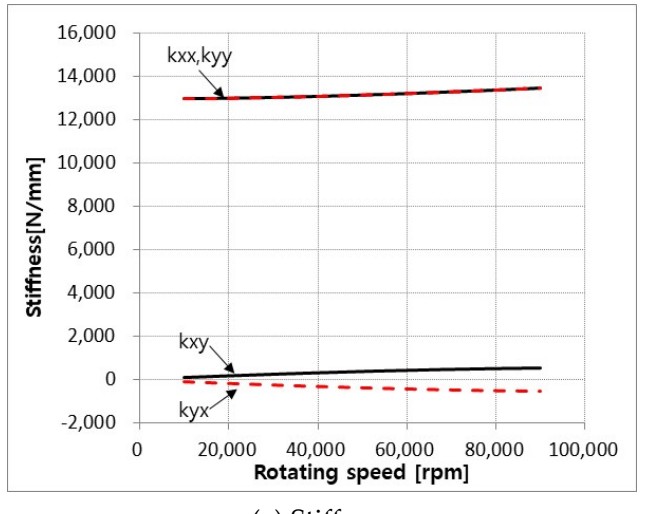

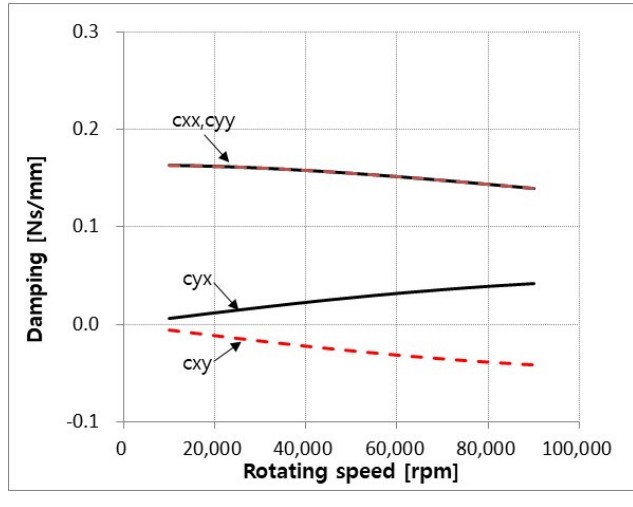

(**a**) Stiffness

(**b**) Damping

**Figure 8.** Stiffness and damping versus rotor speed.

*3.3. Rotordynamic Analysis*

Figure 9 shows the Campbell diagram calculated from the linear rotordynamic analysis. The predicted bearing stiffness and damping, as shown in Figure 8, were applied to the analysis model. Two rigid-body modes were predicted at 36,000 and 40,000 rpm. The predicted rigid-body modes were conical, and the mode with a larger displacement on the compressor side manifested at a lower speed than that with a larger displacement on the turbine side. This is because the compressor impeller was heavier than the turbine impeller. Additionally, the critical speed related to the bending mode was predicted at 110,000 rpm; thus, the separation margin between the rated speed and the critical speed exceeded 45%, indicating that large vibrations would not occur at the rated operation.

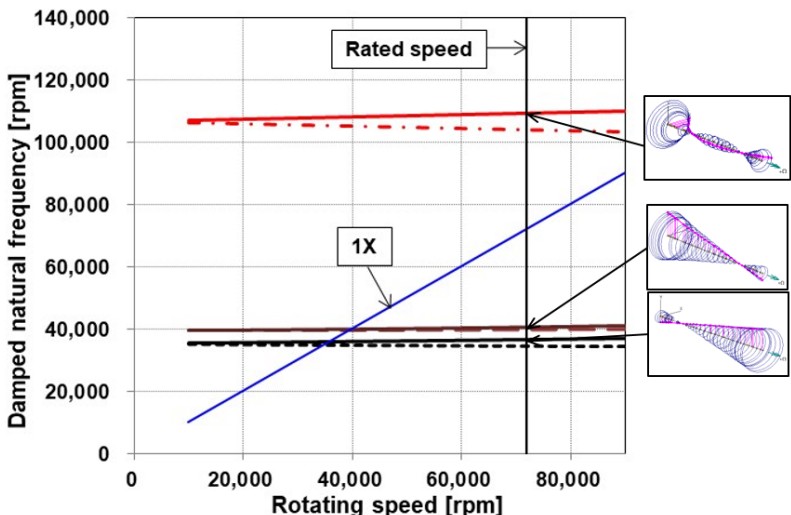

**Figure 9.** Campbell diagram for turbo expander.

Figure 10 plots the predicted rotor orbit obtained through the nonlinear analysis. The analysis was performed by applying an unbalanced mass of ISO 1.0 G at the turbine and compressor impeller nodes with a phase difference of 90°. The phase difference was selected as the measured imbalance of the manufactured prototype rotor. The results presented in the figure represent the steady-state rotor orbit at the bearing location, with the left side indicating the vibration at the compressor side and the right side indicating the vibration at the turbine side for each rotating speed. As shown in Figure 10, the rotor

orbit appeared circular at all rotational speeds. This is because the vertical stiffness of the bearing was identical to its horizontal stiffness, resulting in an isotropic rotor. The amplitude of the rotor was below 1 μm for both the compressor and the turbine bearings at 30,000 rpm; however, as the rotating speed increased to 35,000 rpm, the amplitude at the compressor side first increased, while the amplitude at the turbine side increased as the rotating speed further increased to 40,000 rpm. As the rotating speed increased to 75,000 rpm, the vibration amplitude decreased below 0.5 μm for both the compressor and the turbine sides. In addition, only synchronous vibration was observed up to a rated speed of 75,000 rpm, and no sub-synchronous vibration owing to bearing instability was observed. The maximum amplitude at the compressor side was 36,000 rpm, and that at the turbine side was 40,000 rpm. The speed corresponding to the maximum amplitude correlated well with the critical speed predicted through the linear analysis (Figure 9), indicating that the critical speed could be accurately predicted from the linear rotordynamic analysis. These results suggest that significant vibration issues would not be manifested up to the rated speed if the unbalanced mass is managed under ISO 1.0 G.

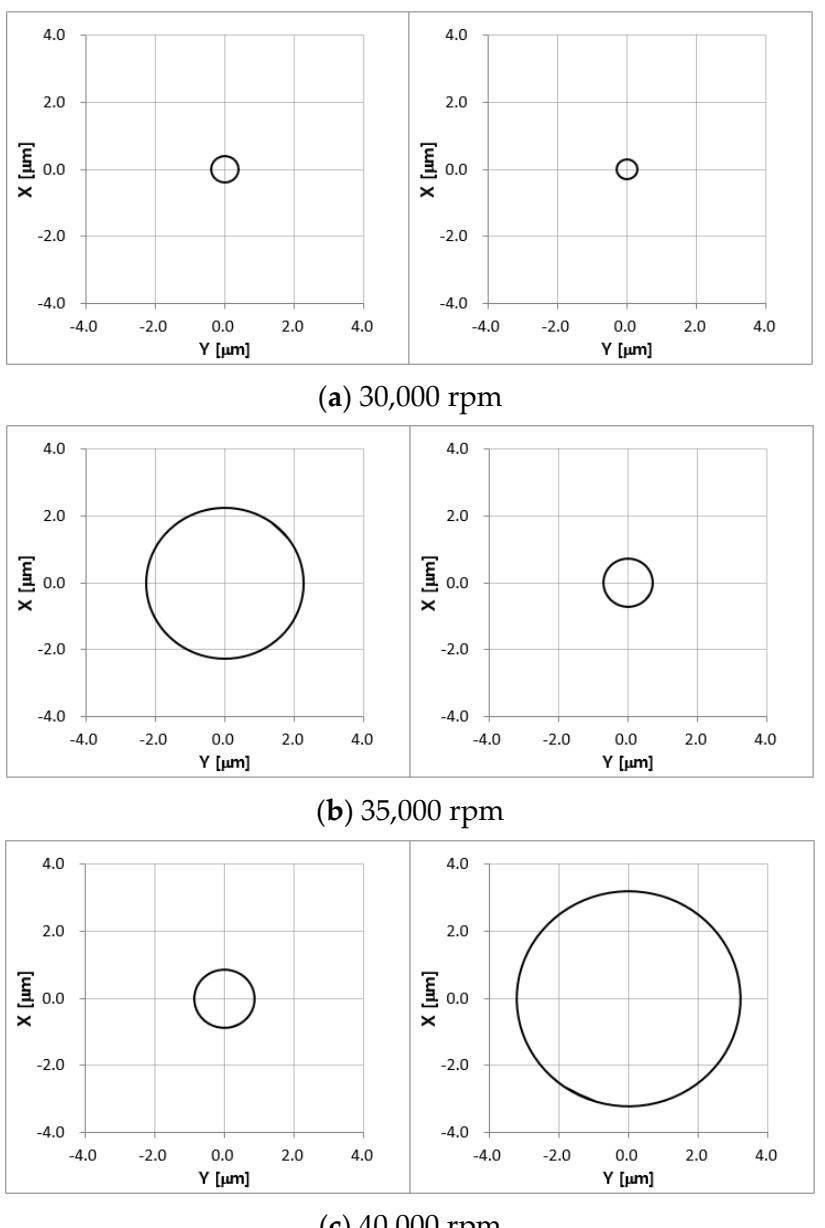

**Figure 10.** *Cont.*

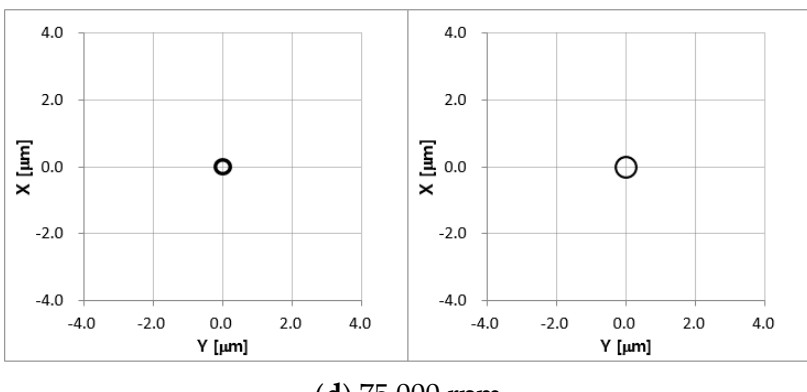

(**d**) 75,000 rpm

**Figure 10.** Shaft orbit for various speeds (**left**: compressor side; **right**: turbine side).

### 3.4. Turbo Expander Operating Test

Figure 11 presents the turbo expander test rig used in the operating test, manufactured by Korea Institute of Machinery Materials in Daejeon, Korea. The shaft was installed vertically, the compressor wheel was located in the upper part, and the turbine wheel was assembled in the lower part of the shaft. To supply pressurized helium to the bearing, six helium containers filled with 110 bar were connected in parallel, as shown in Figure 11b. The helium pressure in the container was reduced to 8.5 bar, which is the bearing supply pressure, through the regulator supplied to the bearing. The operating test was performed by supplying compressed air to the turbine impeller, and the rotational speed was increased by controlling the airflow rate supplied to the turbine. To monitor the shaft vibration during the test, eddy current-type proximity probes were installed near the bearing on the compressor and turbine sides, and the shaft vibration was recorded using a sampling frequency of 5 kHz. In addition, a flowmeter is installed in the helium supply line to measure the flow rate of helium gas in the bearing.

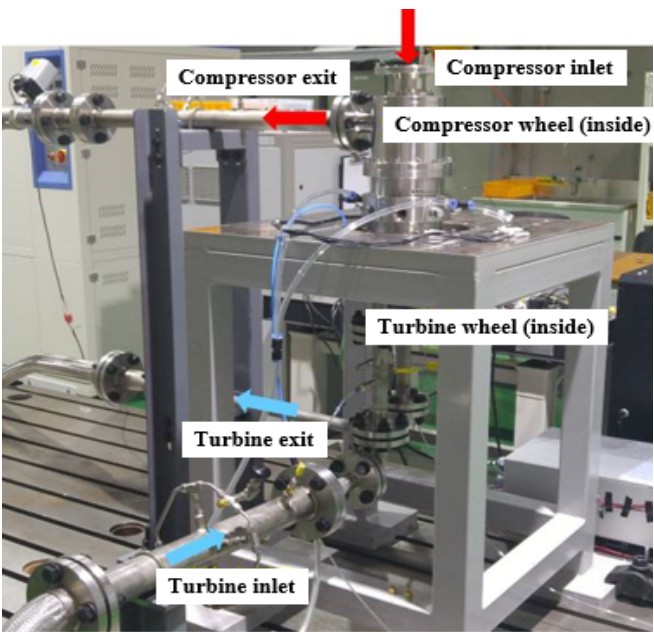

(**a**) Turbo expander

**Figure 11.** *Cont.*

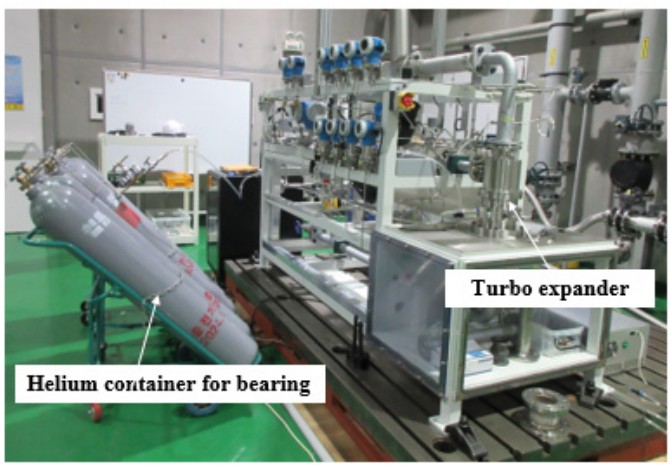

(**b**) Helium supply system

**Figure 11.** Turbo expander test rig.

Figure 12 shows a waterfall plot of the rotor vibration measured during the operational test. The operation was successfully conducted, and no abnormal vibration was observed up to a rated speed of 75,000 rpm. The test results showed that 1× synchronous vibration was dominant for all rotating speeds and sub-synchronous vibration was controlled under the level of 0.1 μm, as shown in the figure. The rotor vibration at the compressor side was maximized at approximately 35,000 rpm, and the maximum amplitude was 2.6 μm. Additionally, the rotor vibration at the turbine side was maximized at approximately 42,000 rpm, and the maximum amplitude was 3 μm. At the rated speed of 75,000 rpm, the rotator vibrations at the compressor side were manifested at the level of 0.1 μm, whereas those manifested at the turbine side measured 1.1 μm. The sub-synchronous vibrations found in the test were associated with the critical speeds presented at 35,000 and 42,000 rpm. Additionally, the local peak observed at 20,000 rpm was considered to be related to the natural frequency of the supporting jig. The measured helium flow rate supplied to the bearing was 2.0 g/s, which was the total flow rate supplied to the two radial bearings and the pair of thrust bearings. The measured flow rate of 2.0 g/s correlated well with the predicted flow rate of 2.04 g/s (1.08 g/s for two radial bearings; 0.96 g/s for thrust bearings). The total flow rate of bearing gas was 2.0 g/s, accounting for 1% of the total flow rate in the process. The bearing gas required for operation was considered a loss within the process. Therefore, further study is required to optimize the design and minimize the consumption of bearing gas.

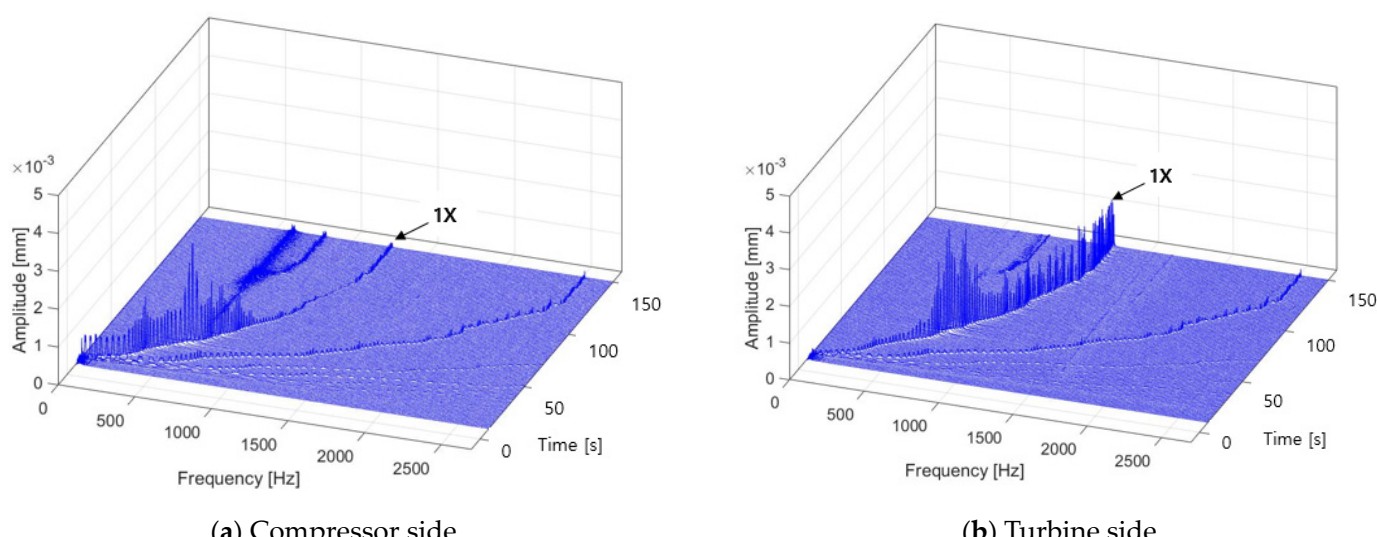

(**a**) Compressor side

(**b**) Turbine side

**Figure 12.** Waterfall plot from measured rotor vibration.

Figure 13 presents a comparison between the rotor vibration predicted using the non-linear orbit simulation and the vibration measured through the operation test. Herein, the predicted amplitude is represented by a solid line, whereas the measured amplitude is represented by symbols. The measured value indicated a vibration amplitude of $1\times$ vibration. Observably, the predicted critical speeds at which the maximum amplitude occurred were similar within 5% of the measured values. In addition, the vibration amplitude at the critical speed was similar to the predicted value, suggesting the credibility of the analytical model.

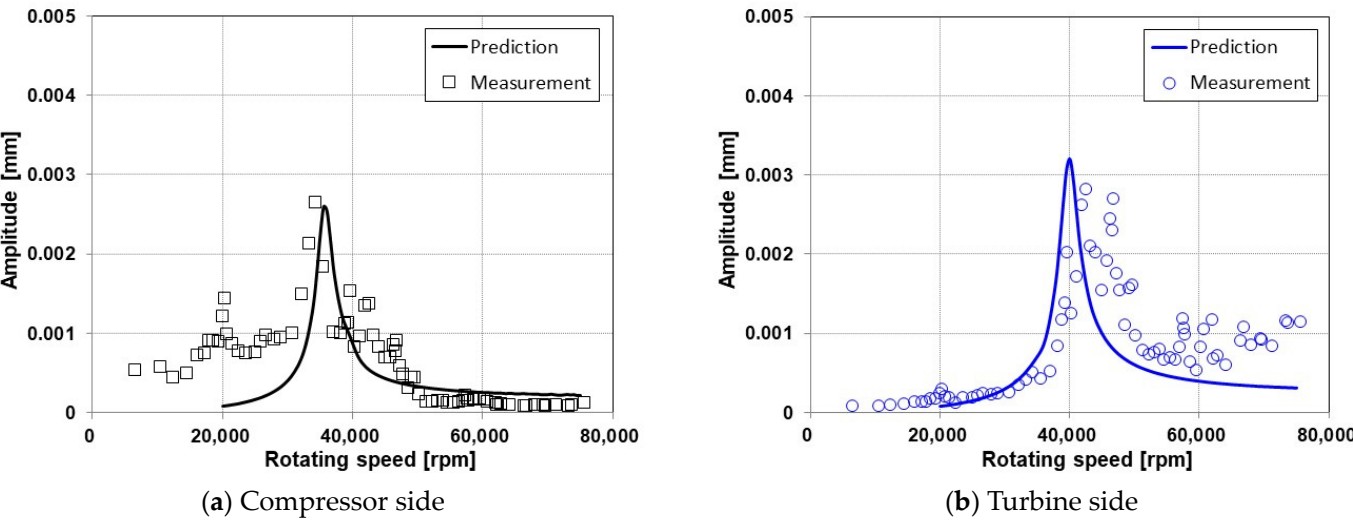

(**a**) Compressor side　　　　　　　(**b**) Turbine side

**Figure 13.** Prediction and measurement of shaft vibration.

## 4. Conclusions

In this study, we analyzed the performance of externally pressurized gas bearings, utilizing helium as a lubricant, to design a turbo expander for the hydrogen liquefaction process. We also predicted the rotordynamic characteristics of the rotor–bearing system. To validate our design, we conducted an operating test and measured the rotor vibration during the test. Our study led to the following conclusions:

1. Under the given operating conditions, we identified a feed parameter that maximizes the stiffness of the radial bearing. The orifice diameter was selected on the basis of these characteristics.

2. The rotordynamic analysis revealed two critical speeds below the rated speed. Moreover, we predicted a separation margin of over 45% between the bending critical speed and the rated speed. These results were consistent with those obtained from the nonlinear orbit simulation.

3. In the operation test, the manufactured turbo expander successfully operated up to the rated speed; the maximum vibration amplitude of the rotor was 3 μm.

4. The critical speed measured in the operation test was within 5% of the values predicted by our analytical model.

In summary, our study demonstrated the feasibility of utilizing externally pressurized gas bearings for the design of a turbo expander for the hydrogen liquefaction process. Furthermore, our rotordynamic analysis and operating test results indicated that the designed turbo expander was reliable and efficient. Future research will focus on optimizing pressurized gas bearings for the turbo expander.

**Author Contributions:** Conceptualization, D.L. and H.L.; methodology, D.L. and H.L.; software, D.L.; validation, D.L., H.L., B.K. and B.J.; formal analysis, B.K.; investigation, B.J.; data curation, H.L.; writing—original draft preparation, D.L.; writing—review and editing, H.L. and J.P.; funding acquisition, H.L. and J.P. All authors have read and agreed to the published version of the manuscript.

**Funding:** This research was partly supported by a grant (1615012665) from the Development of Core Technologies for Commercial Hydrogen Liquefaction Plant Program funded by the Ministry of

Land, Infrastructure, and Transport of the government of South Korea and Korea Institute of Energy Technology Evaluation and Planning (KETEP) grant funded by the Korea government(MOTIE) (No. 20223A10100040, Development of technology for the conversion of retired thermal power generation facilities for synchronous condenser).

**Data Availability Statement:** Data sharing not applicable.

**Conflicts of Interest:** The authors declare no conflict of interest.

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
