# Peer review of "Rotordynamic Analysis and Operating Test of an Externally Pressurized Gas Bearing Turbo Expander for Cryogenic Applications"

_lubricants, doi:10.3390/lubricants11060252_

Round 1
Reviewer 1 Report
1.The formatting of the paper needs to be unified.
2.The meaning of symbols used in the formula is not clearly annotated, such as u1 and u2 in formula 8 of the article. Please carefully check and modify them.
3.The specific labels for radial bearing 1 and radial bearing 2 are not indicated in structural diagram.
4.There are some deviations between the data graph and the text description, such as Figure 6 and Figure10 . Please carefully check and modify them.
5.In the second paragraph of section 3.4, the content described in the text is inconsistent with Figure 12. Please check and modify them.
6.The chapter numbers in the text are incorrect. Please check and modify them.
The level of English writing needs to be improved, and some English sentences in the text are not clear in meaning. Please review and revise the entire text.
Reviewer 2 Report
The paper deserves publication as describes an interesting test rig with also experimental activities and numerical simulations.
Minor comments are inserted in the attached pdf to better clarify some aspects.

Reviewer 3 Report
This paper designs a turbine expander for hydrogen liquefaction process and predicts the dynamics of the rotor-bearing system in this equipment. However, for publication on Lubricants, the reviewer has the following concerns:
1. The form form should use standard scientific paper forms such as table1.
2. The picture type is not uniform, for example, Figure 6-8. The picture style should be consistent with others. Secondly, the coordinates of the picture should be within 4-6.
3. Most of the documents cited are not those in the past five years. Are they of reference value? Is the research a frontier?
4. The innovation of the manuscript is the development of a liquefied hydrogen turbine expansion machine and the prediction of its bearing properties. However, the conclusion part does not highlight the innovation point, and lacks the comparison between the design analysis used in this paper and the existing more advanced methods.
Round 2
Reviewer 3 Report
The manuscript has been well revised, except for one thing: tables in scientific papers should only have three horizontal lines. If this issue is addressed, the manuscript can be considered for acceptance.
Author Response
Thank you for your comment. We have revised the table as you pointed out. Please see page 4.